# Differential Expression and Clinicopathological Significance of HER2, Indoleamine 2,3-Dioxygenase and PD-L1 in Urothelial Carcinoma of the Bladder

**DOI:** 10.3390/jcm9051265

**Published:** 2020-04-27

**Authors:** Donghyun Kim, Jin Man Kim, Jun-Sang Kim, Sup Kim, Kyung-Hee Kim

**Affiliations:** 1Department of Pathology, Chungnam National University School of Medicine, 266 Munhwa Street, Daejeon 35015, Korea; duras3516@cnuh.co.kr (D.K.); jinmank@cnu.ac.kr (J.M.K.); 2Department of Pathology, Chungnam National University Hospital, 282 Munwha-ro, Daejeon 35015, Korea; 3Department of Radiation Oncology, Chungnam National University School of Medicine, 288 Munhwa Street, Daejeon 35015, Korea; k423j@cnu.ac.kr; 4Department of Radiation Oncology, Chungnam National University Hospital, 282 Munwha-ro, Daejeon 35015, Korea; 5Department of Pathology, Chungnam National University Sejong Hospital, 20 Bodeum 7-ro, Sejong-si 30099, Korea

**Keywords:** human epidermal growth factor receptor 2, indoleamine 2,3-dioxygenase, programmed death ligand-1, urothelial carcinoma, urinary bladder, immunotherapy

## Abstract

Purpose: Evasion of the immune system by cancer cells allows for the progression of tumors. Antitumor immunotherapy has shown remarkable effects in a diverse range of cancers. The aim of this study was to determine the clinicopathological significance of human epidermal growth factor receptor 2 (HER2), indoleamine 2,3-dioxygenase (IDO), and programmed death ligand-1 (PD-L1) expression in urothelial carcinoma of the bladder (UCB). Materials and Methods: We retrospectively studied 97 patients with UCB. We performed an immunohistochemical study to measure the expression levels of HER2, IDO, and PD-L1 in UCB tissue from these 97 patients. Results: In all 97 cases, the PD-L1 expression of tumor-infiltrating immune cells (ICs) was significantly correlated with higher pathologic tumor stage (pT). In pT2–pT4 cases (*n* = 69), higher levels of HER2 and IDO expression in invasive tumor cells (TCs) were associated with shorter periods of disease-free survival (DFS). Conclusion: These results imply that the expression of PD-L1 in ICs of the UCB microenvironment is associated with cancer invasion and the expression of HER2 or IDO in the invasive cancer cell and suggestive of the potential for cancer recurrence. We suggest that the expression levels of IDO, HER2, and PD-L1 could be useful as targets in the development of combined cancer immunotherapeutic strategies.

## 1. Introduction

Urothelial carcinoma of the bladder (UCB) remains one of the most common malignant cancers of the genitourinary tract [1]. Among UCB patients, approximately 30% will have muscle invasion at diagnosis, show rapid progression to metastatic disease, and succumb to their disease [2]. Although there are several different treatment regimens, very poor treatment outcomes have been reported in locally advanced and metastatic UCB patients, and this trend has remained unchanged in the last few decades [3,4]. Therefore, further studies are required to better understand the molecular mechanisms of tumor aggressiveness in UCB.

One barrier limiting the efficacy of classic cancer therapies is the interactions of cancer cells with their microenvironment, which ultimately determine whether the primary tumor is eradicated, metastasizes, or establishes dormant micrometastases [5]. Furthermore, the tumor microenvironment can also determine treatment outcome and resistance [6]. Thus, future anticancer treatment strategies should not only act directly on the proliferative processes of transformed cells but also interrupt the crosstalk circuits established by tumor cells with the host microenvironment [7].

Tumor immunogenicity is simply defined as the ability to induce adaptive immune responses [6]. Although most tumors carry particular substances which can induce an immune response, such as antigens or epitopes, the immunogenicity of cancer varies greatly between cancer types. It has been reported that tumor immunogenicity relies on its own antigenicity and several immunomodulatory mechanisms that render tumor cells less sensitive to immune system attack, or create a highly immunosuppressive tumor microenvironment [8]. Classic cancer therapies, such as chemotherapy and radiotherapy, reduce the tumor burden by killing cancer cells. Furthermore, during apoptosis and necrosis, antigens and damage-associated molecular patterns (DAMPs) stimulate an antitumor immune response which induces immunogenic cell death [9,10]. However, cancer cells can escape immune surveillance and progress through modulating immune checkpoint molecules that suppress antitumor immune responses [8].

Considering their high immunogenicity, the expression levels of immune checkpoint-related proteins have been measured and linked to the clinicopathological features and treatment outcomes in UCB. Many studies have reported that expression of immune checkpoint-related proteins, such as programmed cell death 1 (PD-1)/programmed death ligand-1 (PD-L1) and indoleamine 2,3-dioxygenase (IDO) show prognostic significance in UCB [11,12,13]. Additionally, various cancers have responded to treatment with immune checkpoint inhibitors, including UCB [14].

In breast cancer, Filippo et al. highlighted the role of innate and adaptive immune responses in HER2-targeted drugs [15]. This article has prompted investigations into the interaction of immune checkpoint proteins with HER2 targeted therapies. Recently, human epidermal growth factor receptor 2 (HER2) signals have been found to potentially regulate the infiltration of tumor microenvironment immune cells, and to have a role in the expression of PD-L1 in breast and gastric cancers [16,17]. Similarly, increased IDO expression was observed in a subset of HER2+ breast tumors (43.1%), which could be used to develop a combination treatment regimen [18]. These results suggest that immune-escape genes could be used to develop a combination treatment regimen in HER2 overexpression UCB patients. However, the clinical significance of immune checkpoint-related molecules in the context of HER2-positive and -negative UCB have not yet been fully evaluated.

We hypothesized that information on the expression of HER2 and immune-escape genes could be useful in the development of therapeutic strategies. This study aimed to evaluate the expression levels of HER2 and immune-escape genes by immunohistochemistry (IHC) in 97 cases of UCB. Therefore, we first evaluated the influence of immune cell infiltration on UCB survival using the Tumor IMmune Estimation Resource (TIMER) database. Then, to identify immunomodulatory genes, correlations between CD8+ T cell infiltration and candidate genes were analyzed by TIMER. Finally, we evaluated expression levels of HER2, IDO, and PD-L1 by immunohistochemistry (IHC) in 97 cases of UCB. The levels of these three protein expressions were correlated with various clinicopathological characteristics, including patient survival.

## 2. Patients and Methods

### 2.1. Patients and Tissue Samples

This study was approved by the Institutional Review Board of Chungnam National University Hospital (CNUH 2019-10-041). All formalin-fixed paraffin-embedded (FFPE) tissue samples for IHC and clinical data were obtained from the National Biobank of Korea at Chungnam National University Hospital. The requirement for informed consent for the retrospective comparison study was waived because the study was based on immunohistochemical analysis using FFPE tissue.

We conducted a review of the records of 97 patients with UCB between 1999 and 2014 at Chungnam National University Hospital in Daejeon, South Korea. The inclusion criteria were that the FFPE UCB tissues were available, and that the follow-up clinical data were sufficiently detailed. The exclusion criteria were as follows: (1) patients had a previous history of other cancers; (2) patients had received previous curative resection for any urinary tract tumor lesion; (3) patients had received preoperative chemotherapy or radiation therapy; or (4) patients had received any molecular targeted therapy. The tumor, node, and metastasis (TNM) staging and histologic grading for UCB were determined at the time of tumor resection, and were based on the 8th edition of the American Joint Committee on Cancer (AJCC) staging system [19].

The 97 UCB cases included 4 cases of noninvasive papillary urothelial carcinoma, 24 cases of pT1, 40 cases of pT2, 26 cases of pT3, and 3 cases of pT4. The 28 patients who underwent transurethral resection of the bladder (TUR-B) were in the pathologic tumor stage (pT) pTa–pT1; the 69 patients who underwent total or partial cystectomy were pT2–pT4. The histologic type of all 97 cases was conventional urothelial carcinoma. For the 69 cases of pT2–pT4, data were collected regarding their disease-free survival (DFS) and overall survival (OS) periods. Among the 69 cases, 29 patients underwent post-operative radiotherapy (PORT). DFS was determined as the time interval between the date of initial surgical resection and the date of UCB recurrence or metastasis. UCB recurrence or metastasis was determined via imaging and/or histological analysis. OS was defined as from the time of initial surgical resection to the date of death due to any cause. Without confirmation of death, recurrence, or metastasis, OS or DFS time was recorded based on the last known date that the patient was alive. We used representative FFPE whole-tissue samples of 97 UCB cases for immunohistochemistry (IHC).

### 2.2. Immunohistochemical Staining Analysis

Immunohistochemical staining of the FFPE tissue sample of UCB was conducted as previously described [20]. Target Retrieval Solution, pH 9 (catalog #S2368, Dako, Glostrup, Denmark), was used for antigen revitalization. The tissue sections were incubated for 30 min at room temperature with the following primary antibodies: rabbit polyclonal anti-human c-erbB-2 oncoprotein (1:200, catalog #A0485, Dako, Glostrup, Denmark), rabbit polyclonal anti-PD-L1 antibody (1:200, catalog #GTX104763, CD274 molecule, GeneTex, Irvine, CA, USA), mouse monoclonal anti-indoleamine 2,3-dioxygenase antibody, clone 10.1 (1:100, catalog #MAB5412, MERCK, Bellanca, MA, USA), CD8 (Ready-to-Use, catalog #IR623, Dako, Glostrup, Denmark), and CD43 (Ready-to-Use, catalog #IR636, Dako, Glostrup, Denmark).

We only scored HER2, IDO, and PD-L1 IHC stains for invasive urothelial carcinoma cells of 93 invasive UCB cases, while four cases of noninvasive papillary urothelial carcinoma were evaluated for intraepithelial dysplastic urothelial cells. We analyzed the cytoplasmic or cytoplasmic membrane expression of HER2 using the modified DAKO HercepTest ^TM^ Interpretation Manual—Breast Cancer Row version [21] (Staining scored 0, 1+, 2+ and 3+). Staining of 2+ or 3+ was regarded as high expression of HER2. The PD-L1 IHC staining was interpreted using the PD-L1 IHC 22C3 pharmDx Interpretation Manual—Urothelial Carcinoma [22] and VENTANA PD-L1 (SP142) Assay Interpretation Guide for Urothelial Carcinoma [23]. Any convincing partial or complete linear cytoplasmic membrane staining of viable tumor cells (TCs) exceeding 1% of the tumor cell proportion was defined as high expression of TC. Presence of discernible PD-L1, CD43, and CD8 staining of any intensity in the tumor-infiltrating immune cells (ICs) covering ≥1% of the tumor area was regarded as high expression of ICs. For CD43 and CD8, we only scored IHC staining of tumor microenvironment ICs in the muscularis propria of 61 cystectomized UCB cases among 67 cases of pT2–pT4. IDO cytoplasmic expression in TCs was scored using the method described by Allred et al. (score 0–8) [24]. A high expression of IDO was regarded as a median score or above (score ≥5). The results were examined separately and scored by Kim, K-H, and Kim, J-M, who were blinded to the patients’ clinicopathological details. Any discrepancies in the scores were discussed to obtain a consensus.

### 2.3. TIMER Database Analysis

TIMER is a comprehensive resource for systematic analysis of immune infiltrates across diverse cancer types (https://cistrome.shinyapps.io/timer/) [25]. TIMER applies a deconvolution previously published statistical method to infer the abundance of tumor-infiltrating immune cells (TIICs) from gene expression profiles [26]. We investigated the relationship between tumor-infiltrating immune cells and UCB survival outcomes. Additionally, we analyzed the correlation of PDL1, IDO, CTLA4, CCL1, CCL2, and CCR2 expression with the abundance of CD8+ T cells.

### 2.4. Statistical Analyses

The correlations of the clinicopathological parameters with expressions of HER2, IDO, and PD-L1 were evaluated using Pearson’s chi-square test and Fisher’s exact test. The associations between HER2, IDO, PD-L1, CD43 and CD8 proteins were examined by Spearman rank correlation coefficients. Postoperative OS and DFS were determined using Kaplan–Meier survival curves and a log-rank test. The Cox proportional hazards model was applied for univariate and multivariate survival analyses. The mean values of absolute lymphocyte count (ALC), absolute neutrophil count (ANC), and neutrophil to lymphocyte ratio (NLR) were compared for the subgroups with HER2, IDO, PD-L1 (TCs), and PD-L1 (ICs) expression using an unpaired Student’s t-test. Statistical significance was set at *p* < 0.05 (SPSS v.24; SPSS Inc., Chicago, IL, USA).

## 3. Results

### 3.1. Association of Immune Cell Infiltration with Survival and Expression of Immune Escape Genes

Even if there is evidence for the action of various immune cell populations in bladder cancer, a comprehensive landscape of the immune response to UCB and its driving forces is still lacking. Therefore, we tried to identify the correlation between immune cell infiltration of this cancer and survival by using the TIMER (Tumor IMmune Estimation Resource) database. In UCB, only the immune infiltrating level of CD8+ T cells was negatively correlated with survival (Appendix A). These results are in line with the tumorcidial function of CD8+ T in immune cells, which can be mitigated by the immune escape mechanism [27].

It was reported that various molecules may be involved in tumor-induced immune tolerance in UCB [28,29]. Therefore, we evaluated the correlation between CD8+ T cell infiltration of UCB and these molecules by using TIMER. Among various molecules, PD-L1 and IDO1 expression are most highly correlated with CD8+ T cell infiltration in UCB (Appendix A).

### 3.2. Association of Clinicopathological Characteristics with Expression of HER2, IDO and PD-L1

The 97 UCB cases were evaluated using IHC to determine HER2, IDO, and PD-L1 levels. The clinicopathological characteristics of the 97 UCB patients associated with expressions of HER2, IDO, and PD-L1 are presented in Table 1. Most non-neoplastic urothelial epithelial cells or noninvasive urothelial carcinoma cells showed no expression of PD-L1, while HER2 and IDO were generally expressed with mild to moderate intensity in a large majority of reactive urothelial cells or noninvasive intraepithelial urothelial carcinoma cells, while there was no expression of IDO in normal urothelial epithelia. Invasive UCB cancer cells in lamina propria showed a relatively decreased expression of HER2 or IDO in comparison to the expression of reactive or dysplastic intraepithelial urothelial cells (Figure 1). Invasive UCB was scored using IHC stains of deeper invasive cancer lesions, except for intraepithelial lesion. However, the noninvasive papillary urothelial carcinomas were evaluated for intraepithelial dysplastic urothelial cells. Expression of HER2 or IDO in the 97 cases of UCB showed trends of decreased expression in pT2–pT4 compared to pTa–pT1 (*p* = 0.055 and *p* = 0.0007). However, PD-L1 expression of ICs was higher in pT2–pT4 than in pTa–pT1 (*p* = 0.001). HER2 expression in TCs was marginally associated with ALC /μL (*p* = 0.069). IDO expression in TCs was positively correlated with ALC /μL (*p* = 0.030) and was negatively correlated with ANC /μL (*p* = 0.007) and NLR (*p* = 0.050). PD-L1 expression in ICs was positively correlated with ANC /μL (*p* = 0.041) and NLR (*p* = 0.063) (Appendix A).

### 3.3. Correlation Between Expression of HER2, IDO, PD-L1, CD43 and CD8 Measured in Tumor Cells or Immune Cells

The correlation between expression of the five proteins is presented in Table 2. CD43 is one of the major glycoproteins of thymocytes and T lymphocytes, suggesting a negative regulatory role in adaptive immune reactions as one of the positive markers of myeloid-derived suppressor cell phenotyping. The inverse correlation between PD-L1 expression in ICs and IDO expression in TCs was observed (*p* = 0.010). HER2 expression in TC was marginally associated with IDO expression in TCs (*p* = 0.058). There was significant positive correlation between the expression of PD-L1, CD43 and CD8 in ICs. There was a tendency to have a negative feedback phenomenon between the expression of IDO and HER2 in TC and the expression of PD-L1, CD43, and CD8 in ICs.

It has been observed that expression of CD43 and CD8 in tumor microenvironment ICs is generally predominant in the lamina propria rather than the muscle layer. Since CD8 and CD43 expression showed various degrees according to the depth of tumor infiltration, intra-tumoral or contiguous peritumoral ICs in the muscularis propria and deeper layer were evaluated in 61 cases of pT2–pT4 (Figure 2).

### 3.4. Expression of HER2 or IDO May Predict Shorter Disease-Free Survival Period in 69 Cases of pT2–pT4

In pT2–pT4 cases (*n* = 69), we found that expression of HER2 or IDO in TCs was associated with a shorter DFS in both univariate Cox regression analysis (*p* = 0.028 and *p* = 0.048, respectively) (Table 3) and Kaplan–Meier survival curves (*p* = 0.022 and *p* = 0.040, respectively) (Figure 3). The expression of HER2 in TCs was also associated with shorter OS and DFS periods according to multivariate Cox regression analysis for HER2 expression, IDO expression, gender, age, pathologic tumor stage, and radiation therapy after surgery (*p* = 0.031 and *p* = 0.019, respectively) (Table 4). The PD-L1 expression in TCs or ICs showed no correlation with survival outcome (Table 3 and Figure 3), even though the PD-L1 expression of ICs was higher in pT2–pT4 than in pTa–pT1 (*p* = 0.001). The expression of CD43 and CD8 in ICs showed no correlation with survival outcome. In 29 cases of pT2–pT4 with radiation therapy after surgery, the expression of HER2 or IDO in TCs showed an association with shorter DFS in Kaplan–Meier survival curves (*p* = 0.061 and *p* = 0.033) (Figure 4).

## 4. Discussion

In this study, we evaluated the expression of HER2, IDO, and PD-L1 in 97 UCB cases. The three proteins showed a correlation with tumor progression or patient outcome, although they did not show the same trends for clinicopathological correlations. We demonstrated that PD-L1 expression in ICs was significantly higher in pT2–pT4 than in pTa–pT1. Increased HER2 and IDO levels in TCs of 69 pT2–pT4 cases were positively correlated with a shorter DFS period, and could be considered potential factors in poor disease outcomes.

The roles of HER2 and IDO protein in cancer initiation or progression are still poorly understood. The consistent association between the effects of anti-HER2 therapies and immune infiltration has been reported in breast cancer and supports that an anti-tumor immune response can modulate the effect of anti-HER2 therapy [30,31]. In our study, the invasive UCB cancer cells showed a relatively reduced expression of HER2 or IDO in comparison to the expression of reactive or dysplastic intraepithelial urothelial cells. In pTa–pT1 UCBs, the expression of HER2 and IDO increased relative to that of pT2–pT4, apart from that, in pT2–pT4 cases, increased expressions of the two proteins are associated with reduced DFS expression. The altered expression of IDO or HER2 could be interpreted to be a different phase or play a different role for cancer immunoediting to the immune response against noninvasive UCB and invasive UCB [30,32,33]. Our data show a significant positive correlation between the expression of PD-L1, CD43 and CD8 in ICs. It has been observed that there is higher expression of CD43 and CD8 in lamina propria invasion in comparison to muscularis propria invasion. Moreover, there was a tendency to have a reverse correlation between the expression of IDO and HER2 in TCs and the expression of PD-L1, CD43 and CD8 in ICs. Cancer immunoediting describes a complex mechanism between ICs and TCs and has three phases: elimination, equilibrium and escape [34]. In the final escape phase, the expression of IDO in cancer cells inhibits the host immune protection. Paradoxically, IDO is elevated upon various immune molecules of adaptive or innate or tolerogenic immune cells. We speculate that elevated levels of IDO and HER2 in TC may reflect a tumor microenvironment immune reaction. And those immune-evasive transformed cancer cells may reduce IDO expression after down-regulation of immune response with a negative feedback mechanism [30,33,35]. It is predicted that in early cancer development, the expression of IDO or HER2 is upregulated in the majority of cancer cells stimulated by various immune molecules, including IFN-γ, IL-10, IL-27, CTLA4, TGF-β, cyclooxygenase-2 and prostaglandin E2, which are regulated by tumor antigen level or tolerogenic tumor microenvironment [33]. In advanced invasive cancer, the two proteins could be continuously expressed in a relatively reduced number of poorly immunogenic and immune evasive transformed cancer cells, which can lead to a poor prognosis [34]. Therefore, a spatial and periodic variety of cancer immunoediting phase could be in the same tumor mass.

In UCB, HER2 expression status has been evaluated since 1990, when overexpression of HER2 protein was first reported [36]. One study of high-grade UCB (pT2–pT4) ranked the *HER2* gene amplification as the third most significant in terms of associated genetic mutations [37]. Although the first study on the relationship of HER2 expression with clinical outcomes is confounding, a meta-analysis has indicated that its expression is associated with tumor grade, lymph node metastasis, and poor prognosis in UCB [38]. Even so, recent studies have not produced encouraging results for HER2 targeted therapy as a strategy against UCB [39,40,41]. The major scientific reasons for the failure of HER2 targeted therapy are a lack of standardization of HER2 testing and co-expression of other immunomodulatory molecules [42]. To overcome the poor results achieved thus far with anti-HER2 therapy, it is necessary to identify correlations between HER2 and immune checkpoint proteins in UCB. Our study reported that HER2 expression is marginally associated with IDO expression. To the best of our knowledge, this is the first study to correlate HER2 and immunosuppressive molecules in UCB.

Anti-HER2 therapy has revolutionized the treatment of malignant tumors, especially overexpressing breast cancer. Furthermore, with increasing concentrations of anticancer immunotherapy, the connection between HER2 expression and antitumor immunity has emerged as a possible target for combined oncological treatment. The whole-transcriptome profiling of HER2-positive breast carcinomas has revealed a remarkable enrichment in immune pathways [43]. HER2-positive trastuzumab-sensitive breast carcinomas have shown positive associations with chemokines involved in immune cell infiltration of the tumor microenvironment and the expression of PD-1 ligands in tumor cells [16,44]. HER2 expression has recently been found to suppress antiviral defenses and antitumor immunity as a result of HER2 signaling through its intracellular domain, which interferes with cyclic GMP-AMP synthase-stimulator of interferon genes (cGAS-STING) pathway and prevents cancer cell death [45]. Therefore, innate and adaptive immune system responses are increasingly being acknowledged as important regulators of the effects of HER2 targeted therapy [46,47]. Based on previous research, in this study HER2 expression was scored in the cytoplasm as well as the cytoplasmic membrane to include the immune systemic function of the intracellular domain of HER2 signaling. Considering the role of HER2 protein in interfering with antitumor immunity in the cytoplasm, the indications for HER2 targeted therapy are not limited to the cytoplasmic membrane expression of HER2 and we expect that they may also be extended to HER2 protein expression in the cytoplasm of cancer cells.

IDO, also referred to as IDO1, is one of the cytosolic enzymes that catalyzes the initial and rate-limiting steps of tryptophan to kynurenine [33,48]. IDO has been described as having immunosuppressive functions on host immune surveillance of tumor cells, with a focus on its potential immunotherapeutic targets [49]. The role of IDO has been implicated in immune tolerance related to the suppression of T-cell responses such as fetal tolerance, tumor resistance, chronic infections, and autoimmune diseases [50]. One study delineated the action of kynurenine to promote apoptosis in murine bone marrow-derived neutrophils, providing a possible mechanism for increased neutrophil accumulation in IDO-deficient mice [51]. Our results show that IDO expression is correlated with increased ALC and decreased ANC. These findings support previous studies on the immunomodulatory functions of IDO, although its effects or mechanisms in tumor progression remain unclear. IDO expression in TCs showed a negative correlation with ANC and positive correlation with ALC, while the PD-L1 expression in ICs was positively correlated with ANC in the 97 UCB cases.

Recently, phase II and preliminary phase III studies have shown that the application of a PD-L1 inhibitor in metastatic platinum-refractory NSCLC and urothelial cancer resulted in a significant improvement in the response rate and median overall survival [52]. Furthermore, PD-L1 tumor expression has emerged as a biomarker for patient stratification in immunotherapy targeting for the PD-L1/PD-1 pathway, particularly for NSCLC [53]. However, the prognostic impact of this molecule in tumor tissue is still controversial in various cancers, such as NSCLC and head and neck squamous cell carcinoma, because of the high discrepancies between PD-L1 expression and treatment outcomes [54,55]. Some studies have emphasized the significance of a comprehensive evaluation of PD-L1 expression on tumor and immune cells because its expression in immune cells, but not tumor cells, is a favorable prognostic factor for NSCLC and HNSCC [55,56,57]. However, our results show that PD-L1 expression in ICs is a significant poor prognostic factor in UCB.

Radiotherapy induces a host immune response by exposing tumor-specific antigens that make tumor cells detectable by the immune system, promoting the priming and activation of cytotoxic T cells [58]. Furthermore, radiation may have an impact on the tumor microenvironment by facilitating the recruitment and infiltration of immune cells [58,59,60]. Although radiotherapy acts as an in-situ tumor vaccine, it may be insufficient to sustain long-term antitumor immunity, resulting in later relapse [61]. Therefore, there are many studies identifying correlations between molecular regulators of tumor immune escapes and radio-resistance. PD-L1 positive cancer cells have been demonstrated to have a radio-resistant phenotype, inhibiting T cell signaling and T cell-mediated immunogenic cell death [62]. HER2 activation is a potential mechanism that may compromise the outcome of radiotherapy [63,64]. Additionally, in vitro and in vivo experiments blocking PD-L1 and IDO alongside radiation have successfully overcome rebound immune suppression [65,66]. Similarly, our data reveal that the expression of HER2 and IDO are significantly associated with DFS in UCB treated with radiotherapy after surgery (Figure 4).

## 5. Conclusions

The present study is the first to measure the expression levels of IDO, HER2, and PD-L1 and to analyze the correlation between these three proteins and clinicopathological values in UCB. The expression of IDO and HER2 in TCs and PD-L1 in ICs were positively correlated with poor prognostic factors in pT2–pT4 cases, including shorter DFS and OS periods or higher tumor stage. Our results suggest that the expression of IDO, HER2, and PD-L1 are useful as predictive prognostic factors and could potentially be utilized for the development of combined cancer immunotherapeutic strategies.

## Figures and Tables

**Figure 1 jcm-09-01265-f001:**
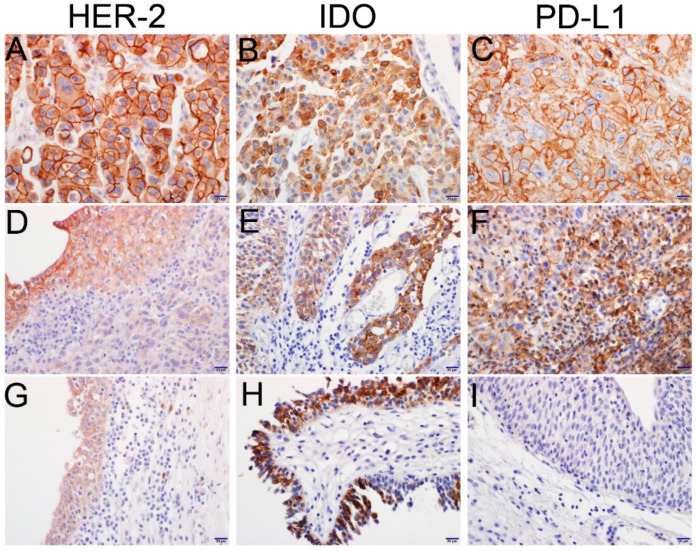
Representative images of HER2, IDO, and PD-L1 immunohistochemical staining in urothelial carcinoma of the bladder (UCB). (**A–C**) Invasive cancer cells with strongly positive expressions of HER2, IDO, and PD-L1. (**D**) Intermediate positive expression of HER2 in low-grade noninvasive urothelial tumor (left upper) and very weakly positive expression of HER2 in invasive cancer cells (right lower). (**E**) Intermediate positive expression of IDO in a low-grade noninvasive urothelial tumor (left) and strongly positive expression of IDO in a high-grade urothelial tumor (right). (**F**) Strongly positive expression of PD-L1 in intra-tumoral immune cells. (**G**) Weakly positive expression of HER2 in reactive urothelial epithelium. (**H**) Strongly positive in situ expression of IDO in urothelial carcinoma. (**I**) Negative expression of PD-L1 in reactive urothelium (scale bar = 20 μm).

**Figure 2 jcm-09-01265-f002:**
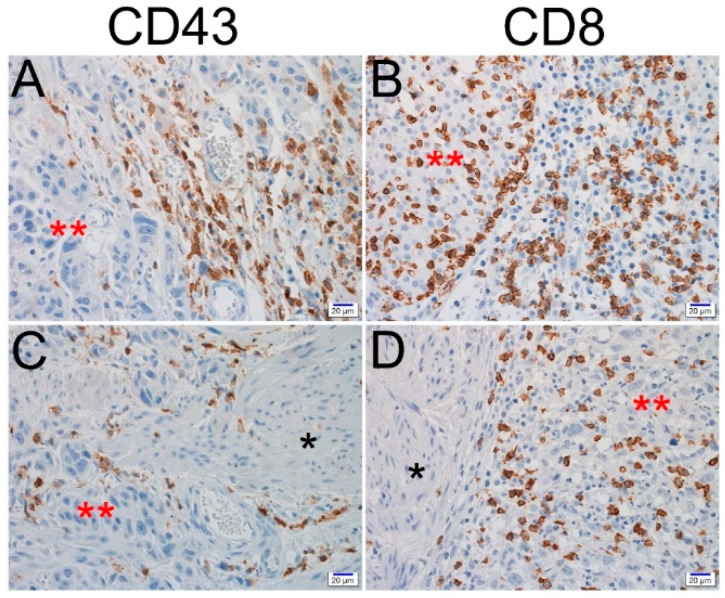
Representative images of CD43 and CD8 immunohistochemical staining in urothelial carcinoma of the bladder (UCB). Positive expression of CD43 and CD8 in intra-tumoral or contiguous peritumoral immune cells of lamina propria invasion (**A**,**B**) and muscularis propria (**C**,**D**) (scale bar = 20 μm; *, muscularis propria; and **, tumor cells).

**Figure 3 jcm-09-01265-f003:**
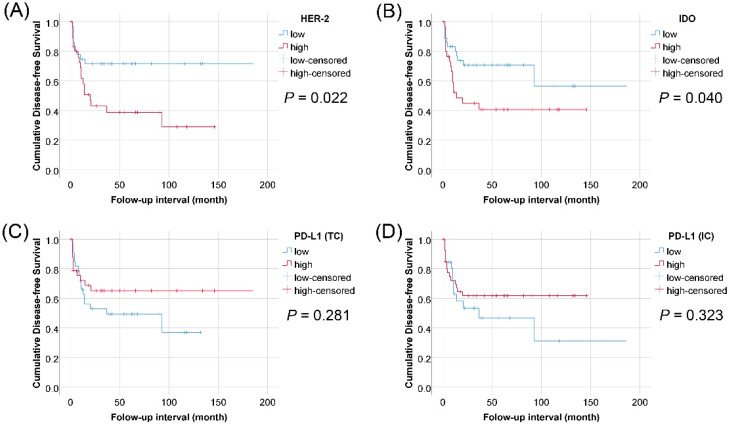
Kaplan–Meier survival curves of disease-free survival in 69 patients with pathologic tumor stage pT2–pT4 urothelial carcinoma of the bladder according to expression of HER2 in tumor cells, IDO in tumor cells, PD-L1 in tumor cells, and PD-L1 in immune cells. (**A**) HER2; (**B**) IDO; (**C**) PD-L1 (TCs); (**D**) PD-L1 (ICs)).

**Figure 4 jcm-09-01265-f004:**
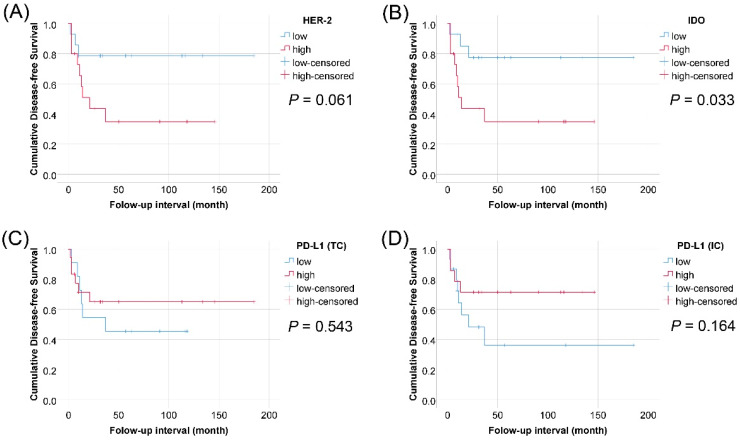
Kaplan–Meier survival curves of disease-free survival in 29 cases with post-operative radiotherapy among 69 patients of pathologic tumor stage pT2–pT4 urothelial carcinoma of the bladder, according to expression of HER2 in tumor cells, IDO in tumor cells, PD-L1 in tumor cells, and PD-L1 in immune cells. (**A**) HER2; (**B**) IDO; (**C**) PD-L1 (TCs); (**D**) PD-L1 (ICs)).

**Table 1 jcm-09-01265-t001:** Correlations of HER2, IDO, and PD-L1 expressions with clinicopathological factors in 97 patients with urothelial carcinoma of the bladder.

Variable	No.	HER2			IDO			PD-L1 (TCs)			PD-L1 (ICs)		
		Low	High	*p* *	Low	High	*p* *	Low	High	*p* *	Low	High	*p* *
Gender		*N* = 46	*N* = 51	0.605	*N* = 45	*N* = 52	0.676	*N* = 52	*N* = 45	0.102	*N* = 48	*N* = 49	0.760
Male	78	38	40		37	41		45	33		38	40	
Female	19	8	11		8	11		7	12		10	9	
Age (years)				0.087			0.164			0.900			0.732
≤65	36	13	23		20	16		19	17		17	19	
>65	61	33	28		25	36		33	28		31	30	
Grade				1.000			0.029			1.000			0.436
low	6	3	3		0	6		3	3		4	2	
high	91	43	48		45	46		49	42		44	47	
Tumor stage				0.055			0.007			0.179			0.001
pTa–pT1	28	9	19		7	21		18	10		21	7	
pT2–pT4	69	37	32		38	31		34	35		27	42	

* Pearson’s chi-square test or Fisher’s exact test.

**Table 2 jcm-09-01265-t002:** Correlations between HER2, IDO, PD-L1, CD43, and CD8 expression according to immunohistochemical staining of urothelial carcinoma of the bladder.

Spearman’s rho		HER2 (TCs)	IDO (TCs)	PD-L1 (TCs)	PD-L1 (ICs)	CD43 (ICs)	CD8 (ICs)
HER2 (TCs)	Correlation coefficient	1.000	0.193	−0.110	−0.155	−0.091	−0.021
	Sig. (2-tailed) *	-	0.058	0.283	0.129	0.485	0.875
	No.	97	97	97	97	61	61
IDO (TCs)	Correlation coefficient	0.193	1.000	−0.171	−0.259 ^*^	−0.247	−0.126
	Sig. (2-tailed) *	0.058	-	0.094	0.010	0.055	0.334
	No.	97	97	97	97	61	61
PD-L1 (TCs)	Correlation coefficient	−0.110	−0.171	1.000	0.383 ^**^	0.242	0.175
	Sig. (2-tailed) *	0.283	0.094	-	0.000	0.060	0.177
	No.	97	97	97	97	61	61
PD-L1 (ICs)	Correlation coefficient	−0.155	−0.259 ^*^	0.383 ^**^	1.000	0.429 ^**^	0.470 ^**^
	Sig. (2-tailed) *	0.129	0.010	0.000	-	0.001	0.000
	No.	97	97	97	97	61	61
CD43 (ICs)	Correlation coefficient	−0.091	−0.247	0.242	0.429 ^**^	1.000	0.608 ^**^
	Sig. (2-tailed) *	0.485	0.055	0.060	0.001	-	0.000
	No.	61	61	61	61	61	61
CD8 (ICs)	Correlation coefficient	−0.021	−0.126	0.175	0.470 ^**^	0.608 ^**^	1.000
	Sig. (2-tailed) *	0.875	0.334	0.177	0.000	0.000	-
	No.	61	61	61	61	61	61

**, Correlation is significant at the 0.01 level (2-tailed); *, Correlation is significant at the 0.05 level (2-tailed); TC, tumor cell; IC, immune cell.

**Table 3 jcm-09-01265-t003:** Univariate analysis of overall survival and disease-free survival in 69 patients with pathologic tumor stage pT2–pT4 urothelial carcinoma of the bladder.

	Overall Survival			Disease-free Survival		
	*P **	HR	95% CI	*P **	HR	95% CI
HER2 expression (TCs)	0.143			0.028		
Low		1 (reference)			1 (reference)	
High		1.792	0.822–3.907		2.381	1.097–5.169
IDO expression (TCs)	0.683			0.048		
Low		1 (reference)			1 (reference)	
High		0.850	0.390–1.852		2.158	1.007–4.622
PD-L1 expression (TCs)	0.854			0.291		
Low		1 (reference)			1 (reference)	
High		1.075	0.498–2.320		0.664	0.311–1.420
PD-L1 expression (ICs)	0.741			0.333		
Low		1 (reference)			1 (reference)	
High		1.146	0.510–2.577		0.692	0.329–1.458
Gender	0.360			0.164		
Male		1 (reference)			1 (reference)	
Female		0.605	0.206–1.774		0.425	0.128–1.417
Age (years)	0.357			0.922		
≤65		1 (reference)			1 (reference)	
>65		1.481	0.643–3.413		0.962	0.444–2.085
Tumor stage	0.016			0.804		
pT2		1 (reference)			1 (reference)	
pT3–pT4		2.639	1.196–5.824		1.100	0.520–2.326
Radiation therapy after surgery	0.395			0.716		
No		1 (reference)			1 (reference)	
Yes		0.706	0.316–1.576		0.870	0.410–1.844

* univariate Cox regression analysis; HR, hazard ratio; CI, confidence interval; TC, tumor cell; IC, immune cell.

**Table 4 jcm-09-01265-t004:** Multivariate analysis of overall survival and disease-free survival in 69 patients with pathologic tumor stage pT2–pT4 urothelial carcinoma of the bladder.

	Overall Survival			Disease-free Survival		
	*P*	HR	95% CI	*P*	HR	95% CI
HER2 expression (TCs)	0.031			0.019		
Low		1 (reference)			1 (reference)	
High		2.501	1.090–5.743		2.729	0.076–6.332
IDO expression (TCs)	0.545			0.101		
Low		1 (reference)			1 (reference)	
High		0.772	0.334–1.786		1.988	0.876–4.514
Gender	0.350			0.054		
Male		1 (reference)			1 (reference)	
Female		0.576	0.181–1.833		0.283	0.078–1.024
Age (years)	0.107			0.858		
≤65		1 (reference)			1 (reference)	
>65		2.036	0.858–4.833		1.079	0.470–2.476
Tumor stage	0.045			0.886		
pT2		1 (reference)			1 (reference)	
pT3–pT4		2.424	1.020–5.760		0.942	0.419–2.118
Radiation therapy after surgery	0.744			0.505		
No		1 (reference)			1 (reference)	
Yes		0.867	0.369–2.039		0.766	0.350–1.675

* multivariate Cox regression analysis; HR, hazard ratio; CI, confidence interval; TC, tumor cell; IC, immune cell.

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
