# Peer review of "Differential Expression and Clinicopathological Significance of HER2, Indoleamine 2,3-Dioxygenase and PD-L1 in Urothelial Carcinoma of the Bladder"

_jcm, 2020, doi:10.3390/jcm9051265_

Round 1

Reviewer 1 Report

This article describes about the prognostic significance and correlation of HER2, IDO, and PD-L1 obtained from surgical specimen. The main theme in the present study is important for the clinical practice and informative. However, a central question is what is the regulative mechanisms of these proteins In addition, the population of this study was heterogenous and the therapeutic regimen was not uniform. And the number of cases was not sufficient.

Author Response

Response to Reviewer 1 Comments

This article describes about the prognostic significance and correlation of HER2, IDO, and PD-L1 obtained from surgical specimen. The main theme in the present study is important for the clinical practice and informative. However, a central question is what is the regulative mechanisms of these proteins In addition, the population of this study was heterogenous and the therapeutic regimen was not uniform. And the number of cases was not sufficient.

It was reported that HER2, IDO and PD-L proteins are involved in an immunomodulatory mechanism that renders tumor cells less sensitive to immune system attack. In the revised manuscript, we evaluated candidate genes associated with CD8+ T cell infiltration in UCB and the relationship between tumor-infiltrating immune cells and urothelial carcinoma of the bladder (UCB) survival outcomes in Table S1 and Figure S1. We described the analyses based on the TIMER (Tumor IMmune Estimation Resource). And we performed immunohistochemical experiments to determine CD8+ T cells and CD43, which is one of positive markers of myeloid-derived suppressor cell phenotyping. We attached the data in the methods, results and discussion sections and in supplementary materials.

The TIMER database and IHC results showed that IDO and PD-L1 proteins are correlated with immune cell infiltration (Table S1 and Table 2). Table S2 shows that there is a correlation between these proteins and systemic inflammatory parameters such as absolute neutrophil count (ANC)/μL, absolute lymphocyte count (ALC) /μL and neutrophil to lymphocyte ratio (NLR). These results reveal the potential regulatory role of these proteins in the antitumor immune response of UCB patients. Our data showed a significant positive correlation between the expression of PD-L1, CD43 and CD8 in tumor microenvironment immune cells. It has been observed that there is higher expression of CD43 and CD8 in lamina propria invasion in comparison to muscularis propria invasion (Figure 2). Moreover, there is a tendency to have a reverse correlation between the expression of IDO and HER2 in tumor cells and the expression of PD-L1, CD43 and CD8 in immune cells. Cancer immunoediting describes a complex mechanism between immune cells and tumor cells and has three phases: elimination, equilibrium and escape [1]. In the final escape phase, the expression of IDO in cancer cells, which depends on anti-tumor immune effector cells, inhibits the host anti-tumor immune response. The elevated levels of IDO and HER2 in tumor cells may reflect a tumor microenvironment immune reaction. And those immune-evasive transformed cancer cells may reduce IDO expression after down-regulation of the immune response with a negative feedback mechanism [2-4]. It is predicted that in early cancer development, the expression of IDO or HER2 to obtain immune evasion is upregulated in many cancer cells, and in advanced invasive cancer, the two proteins are continuously expressed in a relatively reduced number of cancer cells, which can lead to a poor prognosis. Therefore, a spatial and periodic variety of cancer immunoediting phase could be in the same tumor mass.

Table S1. Candidate genes associated with CD8+ T cell infiltration in urothelial carcinoma of the bladder.

Rank

Gene name

purity-corrected partial Spearman’s rho value

P value

1

PD-L1

0.422

3.03E-17

2

IDO

0.297

7.27E-09

3

CTLA4

0.253

9.84E-07

4

CCL2

0.190

2.46E-03

5

CCL1

0.153

3.37-0.3

6

CCR2

0.140

7.18E-03

Table S2. Correlations of HER2, IDO, and PD-L1 expressions with hematologic parameters in 97 patients with urothelial carcinoma of the bladder.

Variable

No.

HER2

IDO

PD-L1 (TCs)

PD-L1 (ICs)

Low

(SD)

High

(SD)

P

Low

(SD)

High

(SD)

P

Low

(SD)

High

(SD)

P

Low

(SD)

High

(SD)

P

ANC /μL

97

5748

(2807)

5533

(3732)

0.752*

6591

(3462)

4808

(2963)

0.007*

5092

(3029)

6262

(3539)

0.083*

4944

(2704)

6312

(3716)

0.041*

ALC /μL

97

1564

(633)

1817

(715)

0.069*

1536

(638)

1837

(701)

0.030*

1711

(674)

1681

(707)

0.832*

1764

(650)

1631

(719)

0.342*

NLR

97

5.82

(10.29)

4.08

(5.15)

0.287*

6.72

(10.56)

3.34

(4.40)

0.050*

3.49

(2.80)

6.53

(11.23)

0.083*

3.38

(3.03)

6.40

(10.7)

0.063*

* Unpaired Student’s t-test ANC, absolute neutrophil count; ALC, absolute lymphocyte count; NLR, neutrophil to lymphocyte ratio.

Figure S1. Kaplan-Meier survival curves comparing the high and low infiltrating levels of CD8+ T cells, CD4+ T cells, macrophages, neutrophils, and dendritic cells in UCB. Infiltrating levels of CD8+ T cells are significantly correlated with poor OS in UCB.

References

  1. Schreiber, R. D.; Old, L. J.; Smyth, M. J., Cancer immunoediting: integrating immunity's roles in cancer suppression and promotion. Science 2011, 331 (6024), 1565-1570, 10.1126/science.1203486.
  2. Hornyak, L.; Dobos, N.; Koncz, G.; Karanyi, Z.; Pall, D.; Szabo, Z.; Halmos, G.; Szekvolgyi, L., The Role of Indoleamine-2,3-Dioxygenase in Cancer Development, Diagnostics, and Therapy. Front Immunol 2018, 9, 151, 10.3389/fimmu.2018.00151.
  3. Spranger, S.; Spaapen, R. M.; Zha, Y.; Williams, J.; Meng, Y.; Ha, T. T.; Gajewski, T. F., Up-regulation of PD-L1, IDO, and T(regs) in the melanoma tumor microenvironment is driven by CD8(+) T cells. Sci Transl Med 2013, 5 (200), 200ra116, 10.1126/scitranslmed.3006504.
  4. Teng, M. W.; Galon, J.; Fridman, W. H.; Smyth, M. J., From mice to humans: developments in cancer immunoediting. J Clin Invest 2015, 125 (9), 3338-3346, 10.1172/JCI80004.

Reviewer 2 Report

The paper is generally well written but there are some major flaws

Line 157: The mean values of absolute lymphocyte count (ALC), absolute neutrophil count (ANC), and neutrophil to lymphocyte ratio (NLR) were compared for the subgroups with HER2, IDO, PD-L1 (TCs), and PD-L1 (ICs) expression using an unpaired Student’s t-test. Statistical significance was set at P < 0.05 (SPSS v.24; SPSS Inc., Chicago, IL, USA).

  • These results have not been presented/discussed in the results section. Instead only a brief mention in discussion section from line 291.

Line 177 - Expression of HER2 or IDO in the 97 cases of UCB showed trends of decreased expression in pT2–pT4 compared to pTa–pT1 (p = 0.055 and p = 0.0007)

Line 206 In pT2–pT4 cases (n = 69), we found that expression of HER2 or IDO in TCs was associated with shorter DFS in both univariate Cox regression analysis (p = 0.028 and p = 0.048, respectively) (Table 3) and Kaplan–Meier survival curves (p = 0.022 and p = 0.040, respectively) (Figure 2).

  • On one hand there is reduced expression of HER2 & IDO in more invasive UCB and on the other hand these are associated with reduced DFS. These two seem to be contradictory and the authors do not provide any explanation for this.

Table 1 is very confusing and needs to be fully reviewed and redone accurately. The percentages brackets are not properly defined as to percent of what and it appears that in lot of places wrong denominators have been used to calculate percentage. E.g. pT2-pT4 row – the percentage for low and high expression should add up to 100% but this is not happening. In ANC & ALC rows the numbers in bracket are more than 100. Therefore these cannot be percentages.

Author Response

Response to Reviewer 2 Comments

The paper is generally well written but there are some major flaws 

Line 157: The mean values of absolute lymphocyte count (ALC), absolute neutrophil count (ANC), and neutrophil to lymphocyte ratio (NLR) were compared for the subgroups with HER2, IDO, PD-L1 (TCs), and PD-L1 (ICs) expression using an unpaired Student’s t-test. Statistical significance was set at P < 0.05 (SPSS v.24; SPSS Inc., Chicago, IL, USA).

  • These results have not been presented/discussed in the results section. Instead only a brief mention in discussion section from line 291.

Thank you for your comments which we have addressed in the result section of the revised manuscript.

 Line 177 - Expression of HER2 or IDO in the 97 cases of UCB showed trends of decreased expression in pT2–pT4 compared to pTa–pT1 (= 0.055 and = 0.0007)

Line 206 In pT2–pT4 cases (n = 69), we found that expression of HER2 or IDO in TCs was associated with shorter DFS in both univariate Cox regression analysis (= 0.028 and = 0.048, respectively) (Table 3) and Kaplan–Meier survival curves (= 0.022 and = 0.040, respectively) (Figure 2).

  • On one hand there is reduced expression of HER2 & IDO in more invasive UCB and on the other hand these are associated with reduced DFS. These two seem to be contradictory and the authors do not provide any explanation for this.

In the discussion section of the revised manuscript, we discussed and described the explanation for the comment as follows.

The roles of HER2 and IDO protein in cancer initiation or progression are still poorly understood. The consistent association between the effects of anti-HER2 therapies and immune infiltration has been reported in breast cancer and supports that an anti-tumor immune response can modulate the effect of anti-HER2 therapy [1-2]. In our study, the invasive UCB cancer cells showed a relatively reduced expression of HER2 or IDO in comparison to the expression of reactive or dysplastic intraepithelial urothelial cells. In pTa–pT1 UCBs, the expression of HER2 and IDO increased relative to that of pT2-pT4; apart from that, in pT2–pT4 cases, increased expressions of the two proteins are associated with reduced DFS expression. The altered expression of IDO or HER2 could be interpreted to be a different phase or play a different role for cancer immunoediting to the immune response against noninvasive UCB and invasive UCB [1, 3-4]. Our data showed a significant positive correlation between the expression of PD-L1, CD43 and CD8 in ICs. It has been observed that there is higher expression of CD43 and CD8 in lamina propria invasion in comparison to muscularis propria invasion. Moreover, there was a tendency to have a reverse correlation between the expression of IDO and HER2 in TCs and the expression of PD-L1, CD43 and CD8 in ICs. Cancer immunoediting describes a complex mechanism between ICs and TCs and has three phases: elimination, equilibrium and escape [5]. In the final escape phase, the expression of IDO in cancer cells, which depends on anti-tumor immune effector cells, inhibits the host anti-tumor immune response. We speculate that elevated levels of IDO and HER2 in TC may reflect a tumor microenvironment immune reaction. And those immune-evasive transformed cancer cells may reduce IDO expression after down-regulation of the immune response with a negative feedback mechanism [1, 4, 6]. It is predicted that in early cancer development, the expression of IDO or HER2 to obtain immune evasion is upregulated in many cancer cells, and in advanced invasive cancer, the two proteins are continuously expressed in a relatively reduced number of cancer cells, which can lead to a poor prognosis. Therefore, a spatial and periodic variety of cancer immunoediting phase could be in the same tumor mass.

 Table 1 is very confusing and needs to be fully reviewed and redone accurately. The percentages brackets are not properly defined as to percent of what and it appears that in lot of places wrong denominators have been used to calculate percentage. E.g. pT2-pT4 row – the percentage for low and high expression should add up to 100% but this is not happening. In ANC & ALC rows the numbers in bracket are more than 100. Therefore these cannot be percentages.

We revised Table 1 and attached Table S1 in the supplementary materials.

References.

  1. Teng, M. W.; Galon, J.; Fridman, W. H.; Smyth, M. J., From mice to humans: developments in cancer immunoediting. J Clin Invest 2015, 125 (9), 3338-3346, 10.1172/JCI80004.
  2. Bianchini, G.; Gianni, L., The immune system and response to HER2-targeted treatment in breast cancer. Lancet Oncol 2014, 15 (2), e58-68, 10.1016/S1470-2045(13)70477-7.
  3. Kim, R.; Emi, M.; Tanabe, K., Cancer immunoediting from immune surveillance to immune escape. Immunology 2007, 121 (1), 1-14, 10.1111/j.1365-2567.2007.02587.x.
  4. Hornyak, L.; Dobos, N.; Koncz, G.; Karanyi, Z.; Pall, D.; Szabo, Z.; Halmos, G.; Szekvolgyi, L., The Role of Indoleamine-2,3-Dioxygenase in Cancer Development, Diagnostics, and Therapy. Front Immunol 2018, 9, 151, 10.3389/fimmu.2018.00151.
  5. Schreiber, R. D.; Old, L. J.; Smyth, M. J., Cancer immunoediting: integrating immunity's roles in cancer suppression and promotion. Science 2011, 331 (6024), 1565-1570, 10.1126/science.1203486.
  6. Spranger, S.; Spaapen, R. M.; Zha, Y.; Williams, J.; Meng, Y.; Ha, T. T.; Gajewski, T. F., Up-regulation of PD-L1, IDO, and T(regs) in the melanoma tumor microenvironment is driven by CD8(+) T cells. Sci Transl Med 2013, 5 (200), 200ra116, 10.1126/scitranslmed.3006504.

Reviewer 3 Report

The authors showed the expression levels of PD-L1, IDO, and HER2 and analyzed the associations among them in UCB. Given that the immunosuppressive role of IDO and HER2, IDO and HER2 may contribute to T cell exhaustion. I suppose that IDO and HER2 might affect other immunosuppressive mechanisms in UCB.

1) I wonder why the authors focused on the expression of IDO and HER2. There are many proteins associated with immunosuppressive tumor microenvironment. Evaluation based on other gene expression database such as TCGA may support their findings.

2) The authors should analyze immune cells by IHC (e.g. CD8+ T cells, regulatory T cells, MDSC, and so on) and evaluate the association of IDO and HER2 expression with immune cell status.

Author Response

Response to Reviewer 3 Comments

The authors showed the expression levels of PD-L1, IDO, and HER2 and analyzed the associations among them in UCB. Given that the immunosuppressive role of IDO and HER2, IDO and HER2 may contribute to T cell exhaustion. I suppose that IDO and HER2 might affect other immunosuppressive mechanisms in UCB.

1) I wonder why the authors focused on the expression of IDO and HER2. There are many proteins associated with immunosuppressive tumor microenvironment. Evaluation based on other gene expression database such as TCGA may support their findings.

Despite evidence for the action of various immune cell populations in urothelial carcinoma of the bladder (UCB), a comprehensive landscape of the immune response to UCB and its impact on survival are still lacking. Therefore, we tried to identify the correlation between the immune cell infiltration of this cancer and survival by using the Tumor IMmune Estimation Resource (TIMER) database. In UCB, only the number of CD8+ T cell infiltrations is negatively correlated with survival. We evaluated candidate genes associated with CD8+ T cell infiltration in UCB and the relationship between tumor-infiltrating immune cells and UCB survival outcomes in Table S1 and Figure S1. We described the analyses based on the TIMER database in the methods and Results sections and in supplementary materials.

Figure S1.

These results are line with the tumorcidial function of CD8+ T in immune cells that can be mitigated by an immune escape mechanism [1]. It was reported that various molecules are involved in tumor-induced immune tolerance in UCB. Therefore, we evaluated the correlation between CD8+ T cell infiltration of this cancer and these molecules by using TIMER (Table S1). The expression of PD-L1 and IDO1 was most highly correlated with CD8+ T cell infiltration in UCB. Due to these results, we focused on the expression of PD-L1 and IDO-1

Gene name

purity-corrected partial Spearman’s rho value

P value

1

PD-L1

0.422

3.03E-17

2

IDO1

0.297

7.27E-09

3

CTLA4

0.253

9.84E-07

4

CCL2

0.190

2.46E-03

5

CCL1

0.153

3.37-0.3

6

CCR2

0.140

7.18E-03

Table S1. Candidate genes associated with CD8+ T cell infiltration in UCB

There have been many trials of human epidermal growth factor receptor 2 (HER2)–targeting agents in HER2-positive UCB patients.  Although there have been early attempts with no definitive clinical efficacy, having the right drugs, or combinations of agents, is often the key. In breast cancer, Filippo et al. highlighted the role of innate and adaptive immune responses in HER2-targeted drugs [2]. This article has prompted investigations into the role of immune checkpoint proteins of HER2-targeted therapies. Recently, HER2 signals have been found to potentially regulate the infiltration of tumor microenvironment immune cells, and to play a role in the increased expression of PD-L1 in breast and gastric cancers [3-4]. Similarly, increased IDO expression was observed in a subset of HER2+ breast tumors (43.1%) [5]. These results suggest that HER2 could be used to develop a synergistic treatment strategy in PD-L1 and IDO overexpression UCB patients. However, the clinical significance of PD-L1 and IDO expression in the context of HER2-positive and -negative UCB has not yet been fully evaluated.

2) The authors should analyze immune cells by IHC (e.g. CD8+ T cells, regulatory T cells, MDSC, and so on) and evaluate the association of IDO and HER2 expression with immune cell status.

We performed IHC experiments to determine CD8+ T cells and CD43, which is one of positive markers of myeloid-derived suppressor cell phenotyping. We attached the data in the results and discuss as follows.

There was a significant positive correlation between the expression of PD-L1, CD43 and CD8 in ICs (Table 2). It has been observed that the expression of CD43 and CD8 in tumor microenvironment ICs is generally predominant in the lamina propria rather than the muscle layer. Since CD8 and CD43 expression showed various degrees according to the depth of tumor infiltration, intra-tumoral or contiguous peritumoral ICs in the muscularis propria and deeper layer were evaluated in 61 cases of pT2-pT4 (Figure 2).

Cancer immunoediting describes a complex mechanism between ICs and TCs and has three phases: elimination, equilibrium and escape [6]. In the final escape phase, the expression of IDO in cancer cells, which depends on anti-tumor immune effector cells, inhibits the host anti-tumor immune response. We speculate that elevated levels of IDO and HER2 in TC may reflect a tumor microenvironment immune reaction. And those immune-evasive transformed cancer cells may reduce IDO expression after down-regulation of the immune response with a negative feedback mechanism [7-9]. It is predicted that in early cancer development, the expression of IDO or HER2 to obtain immune evasion is upregulated in many cancer cells, and in advanced invasive cancer, the two proteins are continuously expressed in a relatively reduced number of cancer cells, which can lead to a poor prognosis. Therefore, a spatial and periodic variety of cancer immunoediting phase could be in the same tumor mass.

References

  1. Rabinovich, G. A.; Gabrilovich, D.; Sotomayor, E. M., Immunosuppressive strategies that are mediated by tumor cells. Annu Rev Immunol 2007, 25, 267-296, 10.1146/annurev.immunol.25.022106.141609.
  2. Bellati, F.; Napoletano, C.; Ruscito, I.; Liberati, M.; Panici, P. B.; Nuti, M., Cellular adaptive immune system plays a crucial role in trastuzumab clinical efficacy. J Clin Oncol 2010, 28 (21), e369-370; author reply e371, 10.1200/JCO.2010.28.6922.
  3. Meng, X.; Huang, Z.; Teng, F.; Xing, L.; Yu, J., Predictive biomarkers in PD-1/PD-L1 checkpoint blockade immunotherapy. Cancer Treat Rev 2015, 41 (10), 868-876, 10.1016/j.ctrv.2015.11.001.
  4. Topalian, S. L.; Taube, J. M.; Anders, R. A.; Pardoll, D. M., Mechanism-driven biomarkers to guide immune checkpoint blockade in cancer therapy. Nat Rev Cancer 2016, 16 (5), 275-287, 10.1038/nrc.2016.36.
  5. Soliman, H.; Rawal, B.; Fulp, J.; Lee, J. H.; Lopez, A.; Bui, M. M.; Khalil, F.; Antonia, S.; Yfantis, H. G.; Lee, D. H.; Dorsey, T. H.; Ambs, S., Analysis of indoleamine 2-3 dioxygenase (IDO1) expression in breast cancer tissue by immunohistochemistry. Cancer Immunol Immunother 2013, 62 (5), 829-837, 10.1007/s00262-013-1393-y.
  6. Schreiber, R. D.; Old, L. J.; Smyth, M. J., Cancer immunoediting: integrating immunity's roles in cancer suppression and promotion. Science 2011, 331 (6024), 1565-1570, 10.1126/science.1203486.
  7. Hornyak, L.; Dobos, N.; Koncz, G.; Karanyi, Z.; Pall, D.; Szabo, Z.; Halmos, G.; Szekvolgyi, L., The Role of Indoleamine-2,3-Dioxygenase in Cancer Development, Diagnostics, and Therapy. Front Immunol 2018, 9, 151, 10.3389/fimmu.2018.00151.
  8. Spranger, S.; Spaapen, R. M.; Zha, Y.; Williams, J.; Meng, Y.; Ha, T. T.; Gajewski, T. F., Up-regulation of PD-L1, IDO, and T(regs) in the melanoma tumor microenvironment is driven by CD8(+) T cells. Sci Transl Med 2013, 5 (200), 200ra116, 10.1126/scitranslmed.3006504.
  9. Teng, M. W.; Galon, J.; Fridman, W. H.; Smyth, M. J., From mice to humans: developments in cancer immunoediting. J Clin Invest 2015, 125 (9), 3338-3346, 10.1172/JCI80004.

Round 2

Reviewer 1 Report

none

Reviewer 3 Report

The authors answered my questions correctly.